# Prediction of the Iron–Sulfur Binding Sites in Proteins Using the Highly Accurate Three-Dimensional Models Calculated by AlphaFold and RoseTTAFold

Béatrice Golinelli-Pimpaneau 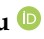

Laboratoire de Chimie des Processus Biologiques, CNRS UMR 8229, Collège de France, Sorbonne Université, 11 Place Marcelin Berthelot, CEDEX 05, 75231 Paris, France; beatrice.golinelli@college-de-france.fr

**Abstract:** AlphaFold and RoseTTAFold are deep learning-based approaches that predict the structure of proteins from their amino acid sequences. Remarkable success has recently been achieved in the prediction accuracy of not only the fold of the target protein but also the position of its amino acid side chains. In this article, I question the accuracy of these methods to predict iron–sulfur binding sites. I analyze three-dimensional models calculated by AlphaFold and RoseTTAFold of Fe–S–dependent enzymes, for which no structure of a homologous protein has been solved experimentally. In all cases, the amino acids that presumably coordinate the cluster were gathered together and facing each other, which led to a quite accurate model of the Fe–S cluster binding site. Yet, cysteine candidates were often involved in intramolecular disulfide bonds, and the number and identity of the protein amino acids that should ligate the cluster were not always clear. The experimental structure determination of the protein with its Fe–S cluster and in complex with substrate/inhibitor/product is still needed to unambiguously visualize the coordination state of the cluster and understand the conformational changes occurring during catalysis.

**Keywords:** AlphaFold; RoseTTAFold; iron–sulfur cluster; Fe–S; Fe–S binding-site; tRNA sulfuration; tRNA thiolation; tRNA modification enzymes; BlsK; structure

## 1. Introduction

### 1.1. AlphaFold and RoseTTAFold

AlphaFold is an artificial intelligence algorithm developed by Deepmind (https://deepmind.com, accessed on 30 November 2021) that accurately predicts the three-dimensional (3D) structure of a protein from its amino acid sequence. AlphaFold relies on multi-sequence alignments of homologs of the protein target and on a deep learning-based neural network to accurately predict the distances between pairs of amino acids. After years of improvements on structure prediction techniques, AlphaFold was recognized in 2020 as a scientific breakthrough in the field of structural biology because of the atomic accuracy of its protein models, even when no homologous structure was known [1,2]. AlphaFold structures had a median backbone of 0.96 Å $rmsd_{95}$ (Cα root mean square deviation at 95% residue coverage) and an all-atom accuracy of 1.5 Å $rmsd_{95}$ compared to experimentally determined structures [1], meaning that it is not only able to predict very accurately the fold of the protein but also to produce highly accurate side chains when the backbone prediction is accurate. In addition, Baker's group developed a related three-track network named RoseTTAFold that integrates information at the one-dimensional (sequence), two-dimensional (distance between pairs of amino acids) and three-dimensional (coordinate) levels to produce structure predictions with accuracies approaching those of DeepMind [3].

The next step is to precisely predict the 3D structures for protein–protein, protein–DNA, protein–RNA, and protein–ligand complexes. Single-chain AlphaFold or RoseTTAFold models represent only one conformational state of a protein. However,

proteins are dynamic systems that adopt various conformations depending on their environment and interacting partners. Thus, Deepmind recently developed the AlphaFold-Multimer algorithm, trained specifically for predicting multimers of known stoichiometry [4]. For heterodimers and homodimers, the interface was successfully predicted in 67% and 69% of the cases, respectively, and high accuracy models were produced in 23% and 34% of the cases, respectively. Baker's group also developed an oligomer structure generation program based on the accurate prediction of interchain contacts [5]. Despite some success by both algorithms in predicting heterodimers and homodimers, both need to be improved, as well as extended to handle higher-order assemblies.

What are the other current limitations of these new structure prediction methods? To date, the algorithms have not been trained or validated to predict the interaction of a protein with a nucleic acid. Moreover, ligands are not included in the AlphaFold and RoseTTAFoldfold structure predictions. One question raised is therefore: how accurately is the cofactor- or ligand-binding site predicted in the 3D model of the protein alone? A successful case has been reported for a well-predicted zinc binding site in a peptidase domain, in which the side chains were in a correct orientation to bind the zinc ion, in complete agreement with the experimentally determined structure [1]. Here, I am interested in Fe–S (iron–sulfur) cluster binding sites and the question whether the 3D model of the apo-protein, as determined by AlphaFold and RoseTTAFold, is sufficiently accurate to predict how the Fe–S cluster would bind to form the holo-protein. During the writing of this article, Wehrspan et al. reported the development of a ligand-search algorithm to identify Fe–S cluster and Zn binding sites in AlphaFold 3D models, which uncovered thousands of novel binding sites for Fe–S clusters [6]. Their results are summarized at the end of this article together with the conclusions of my study.

### 1.2. Does a Given Protein Bind an Fe–S Cluster?

The study of an Fe–S binding protein involves several steps, from the prediction of the existence of the cluster to its spectroscopic, functional, and structural characterization. At first, it is not obvious to know whether a protein binds an Fe–S cluster because the cluster is often very sensitive to oxygen. Thus, this feature may be missed when producing the protein under aerobic conditions. Hints about the presence of a cluster comes from the reddish color of the protein observed after partial purification or from predictions, based on the sequence similarity with another previously characterized Fe–S-binding protein or on characteristic motifs in the sequence. However, although these motifs usually contain cysteines, the most frequent ligands of Fe–S clusters [7], it may be difficult to predict whether a protein binds an Fe-S cluster from its sequence alone because of the variety of the motifs involved [8]. Metalpredator is a web server that has been developed to predict Fe-S binding proteins from protein sequence(s) [9] (http://metalweb.cerm.unifi.it/tools/metalpredator/; 18 December 2021). However, this program was unable to find the Fe–S ligands in several cases investigated in this article (see below).

### 1.3. Biochemical and Structural Characterization of the Fe–S-Binding Protein

Once the existence of an Fe–S cluster is presumed, production of the protein is usually performed under anaerobic conditions, in glove boxes, to prevent the degradation of the cluster by oxygen. When the thus purified protein contains only a low cluster amount, chemical cluster reconstitution is performed to increase the Fe–S content. The cluster is first removed with ethylenediaminetetraacetic acid and dithionite, to prepare the apo-protein, which is then treated with sources of iron (Fe II) and sulfur (provided by cysteine and cysteine desulfurase) in the presence of dithiothreitol (DTT) (reducing conditions). The chemical nature of the cluster (most often [2Fe-4S], [3Fe-4S] or [4Fe-4S]) is determined using various spectroscopies [10]: UV–visible, EPR [11], Mösbauer [12], and resonance Raman [13]. After the biochemical characterization of the protein, it is desirable to have an idea of the structure of the Fe–S binding site to make further hypothesis about the mechanism of the enzyme or function of the protein, which will be challenged by site-

directed mutagenesis and other biochemical methods. It is in particular important to identify the amino acids that coordinate the cluster and know the number of ligands, which impacts the cluster function. Indeed, whereas in a large number of Fe–S binding proteins catalyzing redox reactions, the cluster is bound by four amino acids of the protein, it has been shown that a free coordination site is necessary for a [4Fe-4S] cluster to act as a Lewis acid and bind its substrate [14], as exemplified for aconitase [15,16]. Obtaining the structure of the holo-protein by X-ray crystallography [17] or, when the cluster is sufficiently air-stable, by cryomicroscopy [18], is the Graal, but it is extremely challenging.

In the absence of an experimentally determined structure and given the high accuracy recently achieved in 3D modeling by AlphaFold and RoseTTAFold, it is interesting to question the prediction of the iron–sulfur binding site structures in proteins produced by these programs. Hence, here I analyze several 3D models of enzymes, including tRNA sulfuration enzymes and sulfidases, for which no homologous structure had been solved, to know if the amino acids forming the putative cluster-binding site are already positioned, in the apo-structure, to bind the cluster. For clarity, I use the term "structure" for proteins with experimentally determined structure, deposited at the Protein Data Bank (PDB), and "3D model" when the coordinates are calculated by AlphaFold or RoseTTAFold.

## 2. Results and Discussion

### 2.1. tRNA Sulfuration Enzymes

All tRNA molecules contain post-transcriptional chemical modifications that stabilize the tRNA tertiary structure and are essential for the efficiency and fidelity of the decoding process during genetic translation [19]. Sulfur is present at several positions within tRNAs: U8, C32, U34, A37, and U54 [20–22]. The methylthiolation of the C2 atom of A37 is a redox reaction catalyzed by the S-adenosine-L-methionine radical methylthiotransferases named MiaB [23] and MtaB [24]. In contrast, the sulfuration (or thiolation) reactions at the C2 ($s^2C32$, $s^2U34$, and $s^2U54$) or C4 ($s^4U8$) pyrimidine atoms consist in simple non-redox substitutions of the carbonyl oxygen atom of cytidine or uridine by a sulfur atom, which are catalyzed by tRNA thiocytidine and thiouridine synthetases [25]. Remarkably, several of these enzymes were shown to contain iron–sulfur clusters that are necessary for catalysis [26–31]. Redox sulfuration enzymes use two clusters: one used for generating an S-adenosyl radical and a second, named accessory or additional cluster, the function of which is controversial [32,33]. In the case of non-redox sulfuration enzymes, it was proposed that the [4Fe-4S] cluster, bound by only three ligands, plays the role of a Lewis acid to bind and activate the sulfur atom of the sulfur donor for the tRNA sulfuration reaction [29,34]. Several crystal structures of these latter enzymes have been determined [25]. It can be seen below how information gained from these structures, combined with the biochemical characterization and 3D modeling of homologous proteins with unknown structures, will help to make assumptions about the potential Fe–S binding site of these proteins, and help to plan further experimental validation.

Biosynthetic pathways leading to sulfurated nucleosides were originally classified into two types, depending on the involvement or not of Fe–S-dependent proteins [35]. Among the Fe–S-dependent enzymes, the TtcA/TtuA protein family was shown to be characterized by the existence of both a PP-loop and a central Cys-X-X-Cys (CXXC) motif in the protein sequence [36] (Figure 1). The family was further divided into two distinct groups, based on the presence and location of additional CXXC motifs. TtcA, which catalyzes C32–tRNA sulfuration in bacteria and some archaea, is representative of subtype I, with two central CXXC motifs [36]. TtuA, which catalyzes U54–tRNA sulfuration in archaea and thermophilic bacteria, is representative of subtype II, with only one central CXXC motif and zinc finger sites (each composed of two CXXC/H motifs) at the N- and C-terminal ends [37]. Enzymes, which catalyze U34–tRNA sulfuration in archaea (named NcsA) and the cytosol of eukaryotes (named Ncs6, Ctu1), belong to subtype II [37,38]. In contrast, enzymes, which synthesize $s^2U34$–tRNA in bacteria, named MnmA, represent a different class that possesses a PP-loop motif and only one central D/CXXC motif [39].

TtuA [28,29], TtcA [36,40], NcsA/Ctu1 (Bimai et al., in preparation, PDB code 6SCY), as well as MnmA from *Thermus thermophilus* and *Escherichia coli* [30,31], were shown to use the central CXXC + C motif to bind a [4Fe-4S] cluster. In all of these enzymes, the PP motif (Figure 1) is used to bind ATP to activate the target base.

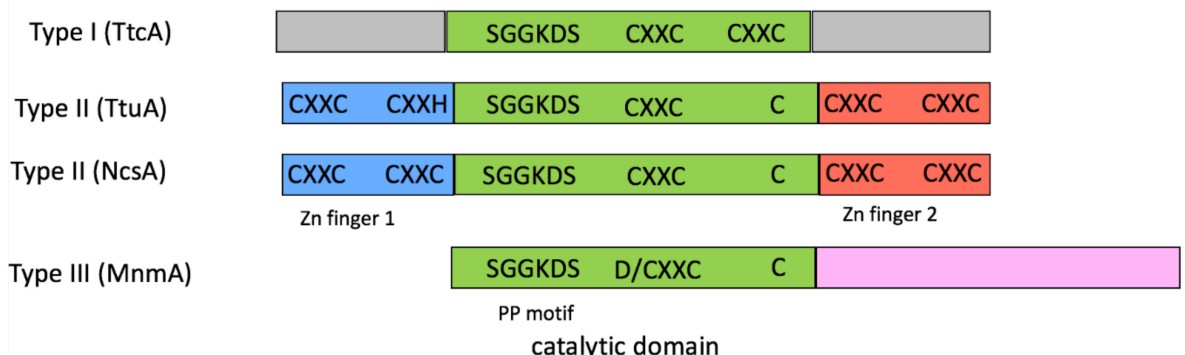

**Figure 1.** Domain organization of several tRNA sulfuration enzymes. TtcA (s$^2$C32), TtuA (s$^2$U54), archaeal NcsA (s$^2$U34), and bacterial MnmA (s$^2$U34) all possess a PP motif (SGGKDS) that binds ATP and a central CXXC + C motif that binds a [4Fe-4S] cluster.

### 2.2. The Cluster Binding Site Is Pre-Formed in the Crystal Structure of the Apo-Form of U54–tRNA Sulfuration Enzyme TtuA

A comparison of previously solved structures of the apo and holo structures of the same Fe–S-binding protein reveals conformational changes occurring upon iron–sulfur cluster binding. TtuA from *T. thermophilus* (TtTtuA) is the only Fe–S-dependent tRNA sulfuration enzyme, for which the crystal structures of both the apo and holo forms have been determined.

The crystal structure of apo-TtTtuA was solved before it was known that it is a Fe–S binding protein [37]. Interestingly, three conserved cysteines, forming the CXXC + C motif (Figure 1), clustered in the active sites: Cys130 and Cys133, which belong to an α-helix, and Cys222 to a flexible loop. Cys130 and Cys222 were involved in an intramolecular disulfide bond, whereas the sulfur atom of Cys133 was positioned 4.5 Å and 5.9 Å away from the sulfur atoms of the two other cysteines (Figure 2, yellow). The mutagenesis of each of the three conserved cysteines, followed by complementation assays of a *T. thermophilus* Δ*ttuA* strain and analysis of the s$^2$U content of the bulk tRNA for each mutant after digestion of the product into nucleosides and HPLC-MS quantification, indicated that all three cysteines were crucial for TtuA catalytic activity [37].

### 2.3. The [4Fe-4S] Cluster of TtuA Is Bound by Three Protein Ligands Only

Next, spectroscopic, enzymatic, and structural studies showed that TtTtuA binds a [4Fe-4S] cluster that is necessary for its tRNA sulfuration activity [28]. The structure of the holo-protein revealed that three iron atoms of the [4Fe-4S] cluster were indeed bound by the three conserved cysteines in the active site, and that the cluster was not coordinated to any other amino acid [28] (Figure 2, green). We reached the same conclusion with the structure of holo-TtuA from *Pyroccocus horikoshii* [29]. Thus, the fourth unique iron atom has a free coordination site, which is presumably used to bind and activate the sulfur donor [29,34]. In the structure of the holo-protein, the intramolecular disulfide bond was broken and the two cysteines involved were displaced by 2.1 Å and 3.5 Å, whereas the sulfur atom of Cys133 moved by only 0.2Å, compared to the apo-structure (Figure 2B). Thus, the binding of the Fe–S cluster triggers a conformational change of the loop that carries Cys222, involved in the disulfide bond, which not only positions Cys222 to properly coordinate the cluster, but also orders the end of the loop (residues 225–228 were not observed in the electron density of the structure of the apo-protein). Therefore, we can conclude that, in this case, the iron–sulfur binding site was almost already set up in the apo-structure, but, as one of

the coordinating cysteines belongs to a flexible loop, it can easily form an intramolecular disulfide bond with one of the other cysteines. The presence of the cluster finalizes the accurate positioning of the cluster ligands and gives rigidity to the binding site.

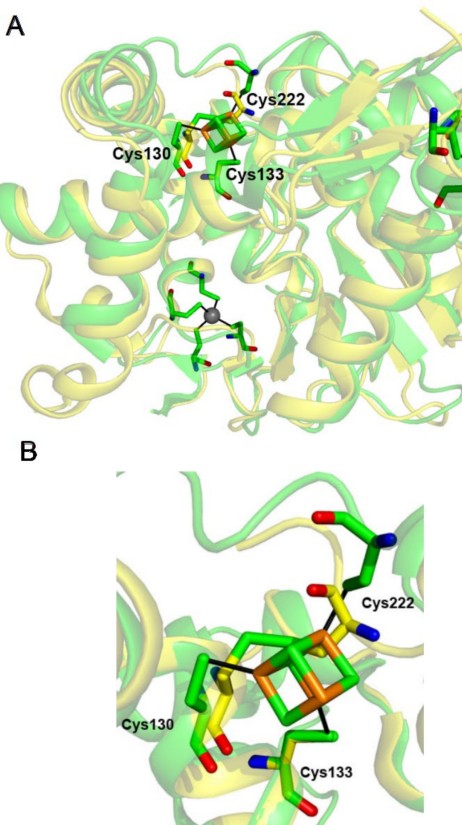

**Figure 2.** Comparison of the crystal structures of apo- (PDB code 3VRH, in yellow) and holo-TtTtuA (PDB code 5B4F, in green). The cysteines of the CXXC + C motif, which coordinate the cluster, are shown in stick representation and labeled. The iron and sulfur atoms of the cluster are shown in orange and green, respectively. (**A**) General view with the zinc ions (grey spheres) and the cysteine/histidine coordinating residues, from the additional CXXC/H motifs, shown in stick representation. (**B**) Comparison of the [4Fe-4S] binding sites of the apo- and holo-TtTtuA structures. In the apo-structure, Cys130 and Cys222 are linked by an intramolecular disulfide bond (2.1 Å) and the sulfur atom of Cys133 is located 5.9 and 4.5 Å away from Cys130 and Cys222, respectively.

*2.4. The Crystal Structure of the U34–tRNA Sulfuration Enzyme MnmA, in the Apo-Form, Suggested That a Cluster Could Bind in the Active Site*

The structure of MnmA from *E. coli* (EcMnmA) in complex with tRNA was solved in three states: the initial binding state, the prereaction state, and, after reaction with ATP, the adenylated state [41]. Like TtuA, the structures of EcMnmA in the first two states showed that two cysteines that are necessary for activity were involved in an intramolecular disulfide bond (Figure 3A, initial binding state in orange).

However, after ATP binding, the disulfide bond was broken (Figure 3A, adenylated state in yellow). The proximity of the two catalytic cysteines, Cys102 and Cys199 [41], with the carboxylate group of Asp99 in the EcMnmA structures, together with the presence of an Fe–S cluster in the Ctu1/Ncs6/NcsA enzymes that catalyze the same $s^2$U34–tRNA formation as MnmA in archaea and in the cytosol of eucaryotes [27], led us to hypothesize that EcMnmA could bind a [4Fe-4S] cluster (Figure 3B). Subsequent spectroscopic, site-directed mutagenesis and enzymatic experiments indicated that EcMnmA binds a [4Fe-4S] cluster coordinated by Asp99, Asp102, and Cys199, which is necessary for catalysis [31]. Accordingly, a [4Fe-4S] cluster can be modeled in the structure of apo-EcMnmA, based on the superposition of EcMnmA with *Methanococcus maripaludis* NcsA (MmNcsA) (Bimai

et al., in preparation, PDB code 6SCY) because the three catalytic residues Asp99, Cys102, and Cys199 of EcMnmA occupy equivalent positions to those of Cys142, Cys145, and Cys233 that bind the [4Fe-4S] cluster in MmNcsA (Figure 3B). Interestingly, like in TtTtuA (Figure 2), Asp99 and Cys102 in EcMnmA, and Cys142 and Cys145 in MmNcsA belong to an α-helix, whereas Cys199 in EcMnmA and Cys233 in MmNcsA belong to a highly flexible loop. Thus, a conformation change of this loop is expected to occur upon Fe–S binding to apo-EcMnmA to move Cys199 away from the two other catalytic residues and position it adequately to bind the cluster. Nevertheless, a crystal structure of holo-EcMnmA is needed to ascertain the mode of cluster binding.

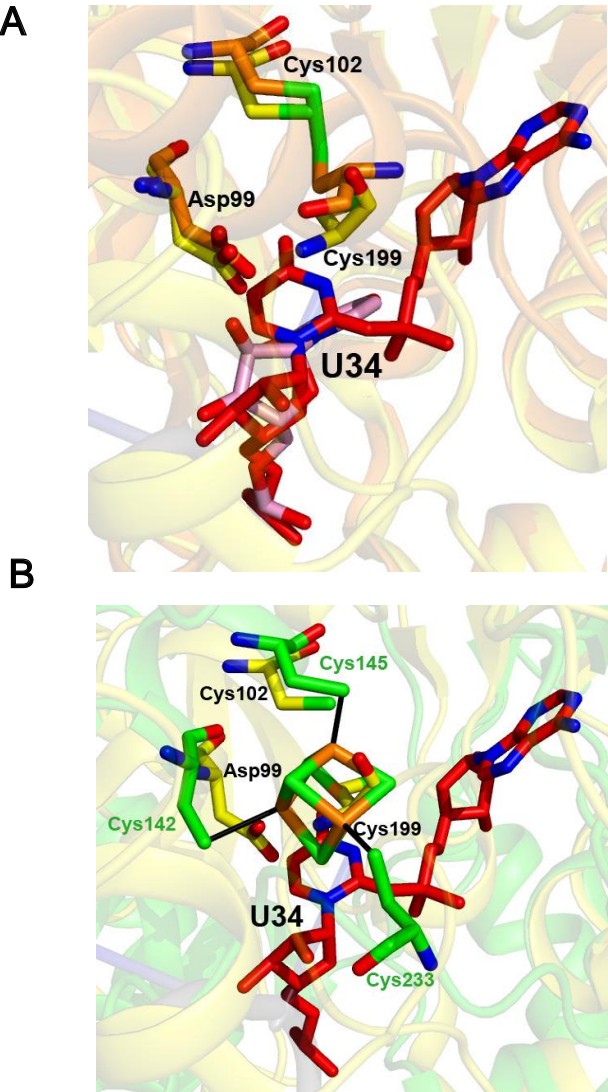

**Figure 3.** The crystal structures of the active site of apo-EcMnmA reveal a potential [4Fe-4S] cluster binding site. (**A**) Superposition of the active sites of tRNA bound EcMnmA in the initial state (molecule B, PDB code 2DER in orange; target uridine 34 in pink) and in the adenylated state (PDB code 2DEU in yellow; adenylated U34 in red), with tRNA[Glu] in grey (rmsd of 0.56 Å for 292 Cα atoms). Cys102 and Cys199 are linked by a disulfide bridge in the initial state. Cys199 belongs to a flexible loop that is not observed in the electron density of molecule A in the initial state. (**B**) Superposition of the catalytic domains of EcMnmA (PDB code 2DEU in yellow) and MmNcsA (PDB code 6SCY in green) using the SSM option in the superpose program of CCP4 (rmsd of 2.37 Å for 123 Cα atoms). The catalytic residues of EcMnmA, Asp99, Cys102 and Cys199 occupy equivalent positions to those of Cys142, Cys145, and Cys233 that bind the [4Fe-4S] cluster in MmNcsA.

*2.5. The Fe–S-Binding Site Is Pre-Formed in the 3D Model of the C32–tRNA Sulfuration Enzyme TtcA, but the Cluster Coordination Is Ambiguous*

By comparison to the TtuA subfamily, the TtcA enzyme subfamily contains a second conserved CXXC motif, in addition to the first central CXXC motif conserved in the whole family (Figure 1). No crystal structure of a TtcA enzyme has been solved yet. The closest homologue with a known structure is TtuA (20% sequence identity), as determined by HHPRED (E-value of $1.3 \times 10^{-24}$ with 231 over 311 residues aligned). In vivo mutational analysis of the two cysteines of the first central CXXC motif of TtcA from *Salmonella enterica* serovar Typhimurium showed that both cysteines are necessary for sulfuration activity [36]. Later, site-directed mutagenesis of each of the four cysteines of the two CXXC motifs was carried out for TtcA from *E. coli* (EcTtcA) [40]. In vivo complementation assays showed that the plasmids with the C122A, C125A, and C213A mutations could not restore synthesis of s$^2$C-tRNA in a $\Delta ttcA$ *E. coli* strain, whereas ones with the C210A mutant could restore ~50% activity. These results suggested that only Cys122, Cys125, and Cys213 are essential for catalysis. As-purified EcTcA contained a low [2Fe-2S] cluster content. Anaerobic purification of the protein did not lead to a fully loaded protein, so anaerobic chemical reconstitution of the cluster was performed. The [4Fe-4S] nature of the cluster of holo-EcTtcA was confirmed by spectroscopic methods. Activity tests based on the quantification of the s$^2$C content of bulk tRNA from the $\Delta ttcA$ strain and the purified as-protein or holo-protein, showed that the [4Fe-4S] cluster is essential for catalysis. To confirm that catalytic Cys122, Cys125, and Cys213 are the ligands of the cluster, the C122A, C125A, and C213 mutants were purified aerobically and characterized by UV–visible spectroscopy, which showed that the as-purified mutants contained a much lower cluster content than the wild-type protein.

In the pathogenic bacterium *Pseudomonas aeruginosa*, expression of the *ttcA* gene increased upon H$_2$O$_2$ exposure, suggesting that TtcA plays a protective role against oxidative stress [42]. Moreover, a *P. aeruginosa* $\Delta ttcA$ strain showed attenuated virulence in a *Drosophila melanogaster* model, indicating that TtcA is required for the successful bacterial infection of the host [42]. To understand whether Fe–S cluster ligation is required for the physiological function of *P. aeruginosa* TtcA (PaTtcA), each cysteine of the two CXXC motifs was mutated to serine, and a complementation assay was performed. H$_2$O$_2$ susceptibility in the $\Delta ttcA$ mutant was not restored by the vectors containing the mutation. This result indicated that the four cysteines of the two CXXC motifs are required for producing a fully functional PaTtcA protein that could play a role in the H$_2$O$_2$-mediated stress response [42]. Similarly, the authors showed that the four cysteines were required for the complete functionality of PaTtcA in bacterial pathogenicity.

Given the important role of the four cysteines of the two CXXC motifs for the in vivo function of PaTtcA and the fact that mutation of one of the four conserved cysteines of EcTtcA, Cys210, led to only a 50% decrease in catalytic activity, the question is raised whether all four cysteines, or only three, coordinate the cluster. Unexpectedly, Metal-predator did not succeed in proposing potential amino acids as candidates for forming the Fe–S-binding site. In the absence of a structure of a TtcA enzyme, we used AlphaFold and RoseTTAFold modeling to understand whether the cluster-binding site is pre-formed in apo-EcTtcA. Moreover, to examine the variability of the positions of the sulfur atoms of the four central cysteines of TtcA proteins and investigate their potential involvement in cluster binding, we superposed the two EcTtcA models with the AlphaFold models of PaTtcA [42] and *Giarda lamblia* TtcA (GlTtcA) [36,42], for which the C32–tRNA sulfuration activity has not been experimentally validated (Figure 4).

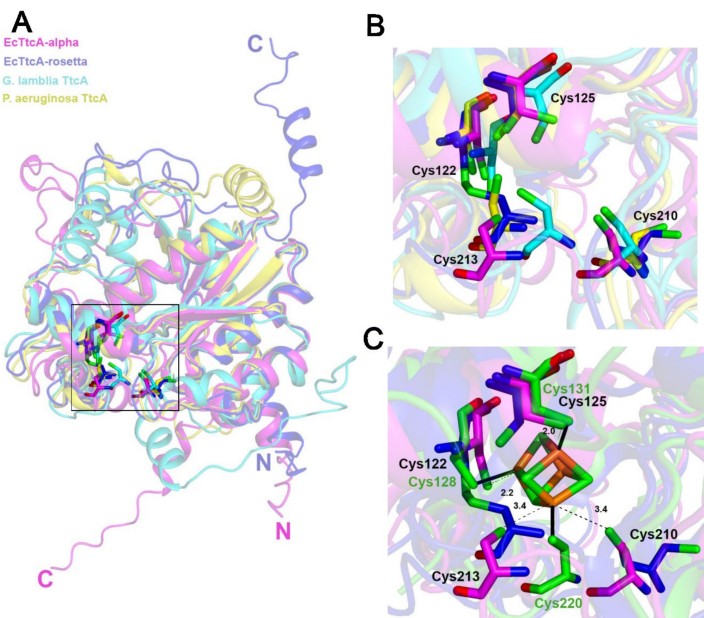

**Figure 4.** Superposition of the RoseTTAFold and AlphaFold 3D models of TtcA from various organisms. EcTtcA, AlphaFold, magenta; EcTtcA, RoseTTAFold, blue; GlTtcA, RoseTTAFold, cyan; PaTtcA, RoseTTAFold, yellow. (**A**) General view. The N- and C-terminal ends of the EcTtcA models are labeled and the cluster binding site is framed. (**B**) Detail of the Fe–S binding site. The four conserved cysteines of EcTtcA are labeled (note that Cys210 and Cys213 were wrongly numbered as Cys219 and Cys222, respectively, in [40]). In the RoseTTAFold model of EcTtcA, Cys122 and Cys213 are linked by an intramolecular disulfide bond. (**C**) Superposition of the Fe–S binding sites of the two EcTtcA models with that of *P. horikoshii* holo-TtuA crystal structure (PDB code 5MKQ). The distances between the sulfur atoms of the cysteines of the EcTtcA AlphaFold model with the nearest iron atoms of the superpositioned [4Fe-4S] cluster of TtuA are indicated, except for the distances of the fourth nonprotein-bonded Fe atom of the TtuA cluster with the Cys122, Cy210, and Cys125 sulfur atoms of EcTtcA, which are 4.3, 4.5, and 4.6 Å away, respectively.

First, it is interesting to note that the predictions by the two algorithms of the structure of the N- and C-terminal ends of EcTtcA are very different, reflecting highly flexible regions (Figure 4A). The core structure is very similar despite several differences located in the Fe–S binding site. Cys122 and Cys213 are linked by an intramolecular disulfide bond in the RoseTTAFold model, but not in the AlphaFold model of EcTtcA (Figure 4B). Moreover, the sulfur atom of Cys210 points towards the inside or outside of the active site, in the AlphaFold and RoseTTAFold models, respectively. Cys122 and Cys125 from the first CXXC motif belong to an α-helix, in contrast to Cys210 and Cys213 from the second CXXC motif that belong to loops. This difference translates into more similar positions, in the four TtcA models, for the cysteines from the first motif, compared to the cysteines from the second motif. The sulfur atoms of the four cysteines in the AlphaFold model of EcTtcA are distanced 4.3 to 7.3 Å from each other (3.5 to 7 Å for the equivalent cysteines in the GlTtcA model, in which all four sulfur atoms point towards the active site). Thus, the models cannot rule out the possibility that Cys210 could be a ligand of the cluster, especially because it belongs to a loop that could undergo a conformational change upon cluster binding. The hypothesis that all four cysteines may bind the [4Fe-4S] cluster was tested by superposing the TtcA models with the structure of holo-TtuA (Figure 4C). In the superposition, all four iron atoms of the TtuA cluster are in close proximity with the sulfur atoms of all four cysteines of EcTtcA, opening up the possibility that the cluster of EcTtcA is bound by all four cysteines. Indeed, in particular, the non-bonded iron atom of the cluster of TtuA is distanced only 4.5 Å from the sulfur atom of Cys210 of EcTtcA (4.3 Å for the equivalent atom in GlTtcA). This potential cluster coordination is intriguing because, as a

non-redox tRNA sulfuration enzyme, TtcA enzymes are expected to have a [4Fe-4S] cluster coordinated by only three ligands [15,16]. However, a conformational change may occur upon substrate-ATP/tRNA/sulfur donor—binding, such that the fourth cysteine ligand is moved away from the catalytic site and can be replaced by the sulfur atom of the sulfur donor. Interestingly, a [4Fe-4S] cluster bound by four cysteines has been observed in the crystal structure of a [4Fe-4S]-dependent serine dehydratase, a non-redox enzyme [43]. In this case, the C-terminal cysteine residue, which is conserved among the family, functions as a fourth ligand to the iron−sulfur cluster producing a "tail in mouth" configuration. This interaction appears to mimic the position that the serine substrate would adopt prior to catalysis. In this regard, extensive sequence alignments produced by RoseTTAFold show that the second CXXC motif is not strictly conserved, as it is sometimes replaced by a CXC motif or even by only one cysteine (Supplementary Figure S1), so that TtcA proteins from some organisms may have their cluster coordinated by three ligands only, in the free state. In summary, crystal structures of TtcA proteins are needed to confirm the cluster ligation, which could fluctuate along the catalytic cycle.

*2.6. The Fe–S-Binding Site Is Pre-Formed in the 3D Model of Cysteine Desulfidase CyuA, but the Cluster Coordination Is Ambiguous*

Desulfidases catalyze the desulfuration of a substrate using a [4Fe-4S] cluster [44,45], which is the inverse reaction of tRNA sulfuration as discussed previously. There are very few proteins from this family that have been characterized and only one structure of one of these enzymes in a holo-form was reported from our group [45].

Cysteine desulfidase is a [4Fe-4S]-dependent enzyme present in many anaerobic bacteria that catalyzes the decomposition of cysteine into pyruvate, ammonia, and hydrogen sulfide [46]. The closest homologue with a known structure is the [4Fe-4S]-dependent serine dehydratase cited above (13% sequence identity) [43], as determined by HHPRED (E-value of $6.3 \times 10^{-43}$ with 359 over 388 residues aligned). Cysteine desufidase from *Methanococcus jannaschii* (MjCyuA) were overexpressed in *E. coli* [44]. The aerobically purified recombinant enzyme, which contained 2.7 mol of Fe per subunit and had an absorbance spectrum characteristic of a [3Fe-4S] cluster, was inactive but could be activated in the assay mixture, by conversion of the [3Fe-4S] cluster into the active [4Fe-4S] cluster with ferrous ions and DTT. To test the functional importance of the four conserved cysteines (Cys25, Cys282, Cys322, and Cys329) of MjCyuA, the C25A, C282A, and C322A–C329A mutants were produced. The C25A, C282A, and C322A–C329A mutants possessed 0.4%, 5%, and no catalytic activity, respectively. Tchong et al. suggested that the residual activity of the C282A mutant (5%) might result from ligand rearranging, with Cys284 serving as the fourth ligand of the cluster [44]. Treatment of the enzyme with N-ethylmaleimide and iodoacetamide led to the inactivation and alkylation of Cys25, which suggested that Cys25 serves as a base to abstract the Cα hydrogen of the L-cysteine substrate in the first step of the reaction. Altogether, it was concluded from sequence alignment, site-directed mutagenesis, and biochemical assays that Cys282, Cys322, and Cys329 are the likely ligands of the [4Fe-4S] cluster. Intriguingly, Metalpredator predicts that Cys284 (and not Cys282), Cys322, and Cys329 coordinate the Fe–S cluster of MjCyuA. To know if Cys282 or Cys284 is a cluster ligand and if the cluster is coordinated by three or four ligands, I calculated the 3D models of MjCyuA (Figure 5).

Both the AlphaFold and RoseTTAFold models are highly similar, with only the first and last amino acids showing some disorder. The structure of MjCyuA is organized into two tightly packed domains: the N-terminal domain and the C-terminal catalytic domain that contains the cluster binding site. Despite a very similar fold, some differences are observed in the Fe–S binding site of the two models (Figure 4B). Cys284 and Cys329, which lie within helical segments, occupy very similar positions, whereas Cys282 and Cys322, which both belong to flexible loops, are linked by an intramolecular disulfide bond in only the RoseTTAFold model. In both models, Cys25 is located farthest away (>7 Å) from the four other cysteines in a flexible loop, in agreement with it playing no role in cluster

coordination but rather a catalytic function. The sulfur atoms of the four other cysteines in the catalytic domain are 5.4 to 6.4 Å away from each other in the AlphaFold model (Figure 5C), indicating that they could all participate in the coordination of the [4Fe-4S] cluster. As both Cys282 and Cys284 are putative cluster ligands, the conservation of these residues during evolution was questioned. An analysis of the multisequence alignment produced by RoseTTAFold (Supplementary Figure S2) confirms the finding by Tchong et al. that Cys282 is strictly conserved, and not Cys284. Thus, the coordination of the cluster by Cys282 is more likely than by Cys284. In any case, the ambiguous coordination of the cluster of MjCyuA is an interesting illustration of the limits of the modeling methods.

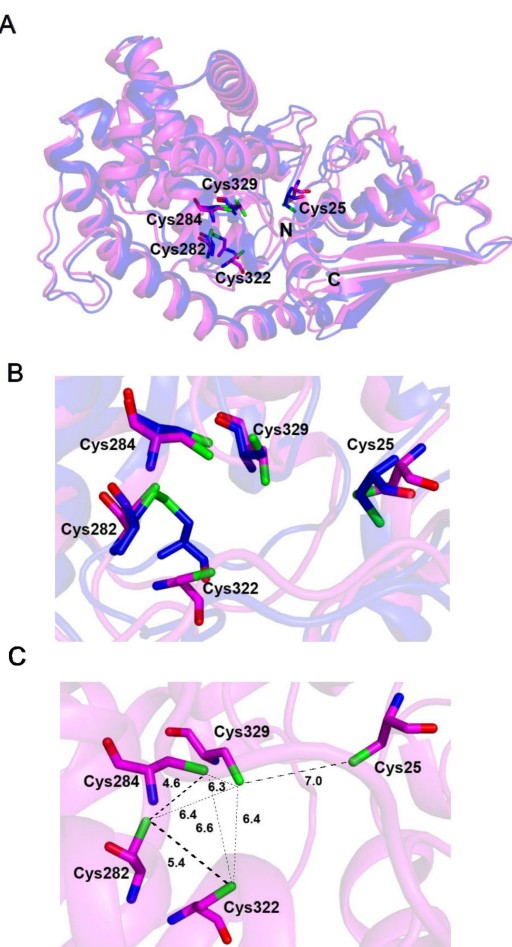

**Figure 5.** MjCyuA 3D models. (**A**) Overall view of the superposition of the RoseTTAFold (blue) and AlphaFold (magenta) 3D models of MjCyuA. The N- and C-terminals are labeled. (**B**) Enlarged view of the Fe–S binding sites. In the RoseTTAFold model, Cys282 and Cys322, are linked by an intramolecular disulfide bond. (**C**) Detail of the Fe–S binding site of the AlphaFold model. The main distances between the sulfur atoms of the five conserved cysteines are indicated (in Å).

Tchong et al. proposed a mechanism in which the [4Fe-4S] cluster is bound by three protein ligands only, so that the fourth nonprotein-bonded iron atom can act as a Lewis acid and bind the sulfur atom of the L-cysteine substrate [44]. However, even if the four cysteines coordinate the [4Fe-4S] cluster of MjCyuA in its empty form (no substrate/inhibitor/product ligand), it is possible that, in the presence of the L-cysteine substrate, one of the cysteines in the flexible loops moves away from the cluster and is replaced by the sulfur atom of the substrate. In that case, the role of the fourth cysteine of the protein would be to protect the cluster from degradation by oxygen in the holo-MjCyuA alone form. As discussed above, such a [4Fe-4S] cluster bound to four cysteines from the protein was observed for a serine dehydratase [43]. Several crystal structures of holo-MjCyuA are needed to determine the

coordination state of the resting enzyme and observe the potential conformational changes in Fe–S coordination upon substrate or inhibitor binding.

### 2.7. Blsk, a [3Fe-4S]-Dependent Enzyme Involved in Amide Bond Formation with an Unkown Mechanism

BlsK catalyzes the leucyl transfer from leucyl-transfer RNA to the β-amino group on the arginine side chain of an intermediate in the biosynthesis of blasticidin S, a peptidyl nucleoside antibiotic. The enzyme from *Streptomyces griseochromogenes* was recently purified and characterized [47]. It was shown to contain a [3Fe-4S] cluster necessary for its catalytic function. The exact role of the cluster in the amide bond formation catalyzed by BlsK, a non-redox reaction, remains unknown. Site-directed mutagenesis of each cysteine of BlsK to serine, followed by expression, purification, and spectroscopic characterization of the mutant proteins, revealed that mutations of Cys236, Cys253, or Cys259 led to the loss of the [3Fe-4S] cluster [47]. This suggested that the cluster in BlsK is coordinated by these three cysteines. Metalpredator did not succeed in finding these cysteines as candidates for cluster binding. This is likely due to the fact that Blsk does not show any similarity to any known functional protein. Indeed, HHpred did not detect any close homolog to BlsK in the PDB. Therefore, BlsK represents a new type of an amino–acyl-tRNA dependent enzyme.

I used AlphaFold and RoseTTAFold to predict the fold of BlsK (Figure 6A) and examined the presumed Fe–S binding site (Figure 6B,C).

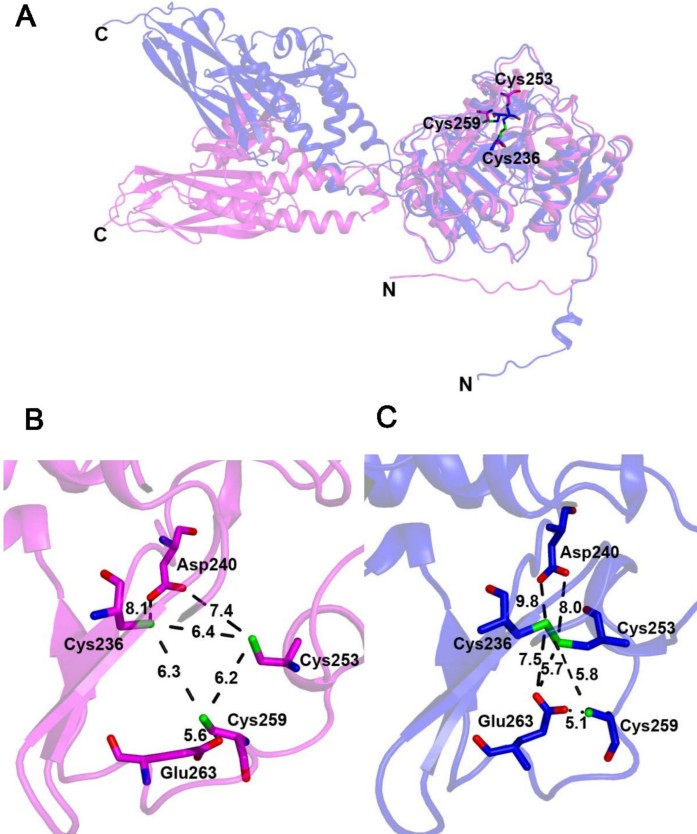

**Figure 6.** (**A**) Superposition of the RoseTTAFold and AlphaFold 3D models of BlsK. The C-terminal domain of the alpha-Fold (in magenta) and RoseTTAFold (in blue) models of BlsK are superpositioned. The cysteines that most likely coordinate the cluster are labeled and shown as sticks. (**B**) Detail of the Fe–S binding site of the AlphaFold model. The distances between the sulfur atoms of the three cysteines that were suggested to bind the cluster are indicated (in Å). Additional distances are indicated with two other residues in the active site, Glu263 and Asp240, which could play a role in catalysis or cluster binding. (**C**) Detail of the Fe–S binding site of the RoseTTAFold model. Cys236 and Cys253 form an intramolecular disulfide bond.

The protein is organized into two domains linked by a very long linker (residues 358–378), which results in flexibility of the N-terminal and C-terminal domains relative to each other, as observed by the superposition of the C-terminal domains of the 3D models calculated with AlphaFold and RoseTTAFold (Figure 5A). Moreover, the N-terminal tail appears to be highly flexible. The Fe–S binding site is located in the N-terminal domain, with Cys236, Cys253, or Cys259 being located close to each other. Indeed, in the AlphaFold model, the sulfur atoms of these three cysteines are 6.2–6.4 Å away from each other, confirming that they are most likely the ligands of the cluster (Figure 6B). Interestingly, the carboxylate side chains of Asp240 and Glu263 are located near these three cysteines, indicating that they could play a role in catalysis or provide a fourth protein ligand to the cluster [48]. A comparison of the Fe–S binding sites of the AlphaFold and RoseTTAFold models (Figure 5B,C) highlights some differences: in the RoseTTAFold model only, Cys236 and Cys253 are linked by a disulfide bridge and the Glu263 side chain is oriented towards the active site, in close proximity to all sulfur atoms of Cys236, Cys253, or Cys259 (5.1–7.5 Å). These models encourage further investigation into the mechanism of BlsK to confirm that the [3Fe-4S] cluster, and not the [4Fe-4S] cluster, is the catalytic form, to assess that it is bound by three cysteines only, and to propose a mechanism by which it could play a catalytic role.

## 3. Methods

HHPRED [49] was used to search for the closest homologues of the protein of interest, whose structures had been experimentally determined.

The RoseTTAFold models were calculated using the RoseTTAFold option and providing the amino acid sequence to the server https://robetta.bakerlab.org/submit.php, accessed on 30 November 2021. The confidence score (1 good; 0 bad) of the model corresponds to the average pairwise Template Modeling (TM) score of the top 10 RoseTTAFold scoring models. To analyze conservation of amino acids, the multi-alignment sequence file t000_.ss2.a3m was given as input to ClustalW [50] without using the seeded guide trees and HMM profile-profile options, and visualized with EsPRIPT [51].

The AlphaFold models were calculated with the Google Colab platform and AlphaFold2_advanced option https://colab.research.google.com/github/sokrypton/ColabFold/blob/main/beta/AlphaFold2_advanced.ipynb#scrollTo=ITcPnLkLuDDE, accessed on 30 November 2021, [52] that does not use templates (homologous structures), and refined using the Amber-relax option to enhance the accuracy of the side chains geometry. The predicted local-distance difference test (pLDDT) confidence values (higher = better) are indicated in the B-factor column.

In both modeling algorithms, only the best model among the five best given by default was examined in detail and represented in the Figures. The RoseTTAFold confidence score and AlphaFold pLDDT and TM scores for the proteins studied are indicated in Table 1.

**Table 1.** Prediction of the 3D model of Fe–S-dependent enzymes discussed in this paper that have no close homologs in the PDB.

| Protein | RoseTTAFold (Confidence Score) | AlphaFold (pLDDT) | AlphaFold TM Score |
|---|---|---|---|
| EcTtcA | 0.74 | 85.8 | 0.83 |
| PaTtcA | 0.79 | - | - |
| GlTtcA | 0.71 | - | - |
| MjCyuA | 0.82 | 93.9 | 0.92 |
| BlsK | 0.80 | 92.9 | 0.86 |

## 4. Conclusions

In this article, I provided several examples of 3D model predictions by RoseTTAFold and AlphaFold of Fe–S-dependent enzymes, for which there was no close homolog in the PDB. In all cases, the cluster-coordinating cysteines were in close proximity to each other,

defining quite accurately the cluster binding site. More generally, Wehrspan et al. very recently showed that, in a large number of cases out of the 362,198 AlphaFold models from the 21 complete AlphaFold proteomes screened by their ligand-search algorithm, the predicted Fe–S and Zn binding sites were sufficiently well-formed to bind the ligands and, more importantly, that the binding site could be unambiguously assigned to the [4Fe-4S], [2Fe-2S], or Zn type [6]. Interestingly, in our study, as observed in the crystal structures of several apo-proteins (TtuA, MnmA), two of the cysteines formed a disulfide bond in the RoseTTAFold models, but not in the AlphaFold models. Yet, a survey of a much larger number of cases than presented here indicated that AlphaFold also occasionally builds erroneous disulfide bonds between cysteines that should instead coordinate a ligand [6]. These findings indicate that the prediction of the presence or absence of disulfide bonds between closely located cysteines remains a challenge, which needs to be better taken into account in the future by the AlphaFold and RoseTTAFold algorithms. Altogether, the models confirmed that the cluster-binding cysteines are relatively flexible in the absence of the cluster, and indicated that only small rearrangements of the active site amino acids are needed to accurately position the cysteines upon cluster binding.

Interestingly, Wehrspan et al. showed that the AlphaFold 3D models can state the chemical nature of the cluster ([2Fe-2S] or [4Fe-4S]), which, however, remains to be confirmed experimentally [6]. Moreover, as shown here, the models can also help anticipate the number of ligands of the cluster and wonder if the cluster may be bound by a nonconventional ligand (such as aspartate, glutamate, arginine, or serine) in addition to cysteines. This, in turn, would help create hypotheses about the catalytic mechanism in the case of enzymes. In particular, in the case of non-redox enzymes, it is possible that the fourth ligand of the cluster is used to protect the [4Fe-4S] cluster in the absence of substrates and be released upon substrate binding. In addition to the 3D models, RoseTTAFold and AlphaFold provide for each protein target, a multi-alignment of a huge number of sequences, which allows the analysis of the evolutionary conservation of a given amino acid, and hence, the prediction of a conserved function.

Finally, after 3D modeling, the presumed ligands and function of the cluster need to be tested experimentally. Indeed, they may sometimes be validated only by obtaining the crystal structure of the holo-protein. In this regard, another great power of RoseTTAFold and AlphaFold modeling is, first, to predict which protein construct or protein from which organism will be more structured and more likely to crystallize, and second, to provide a model that can be used in molecular replacement methods to solve the phasing problem and determine the crystal structure.

**Supplementary Materials:** The following are available online at https://www.mdpi.com/article/10.3390/inorganics10010002/s1, Figure S1: Multi-sequence alignment produced by RoseTTAFold of the two regions with the conserved CXXC motifs of TtcA proteins. The first sequence and numbering correspond to EcTtcA; Figure S2: Multi-sequence alignment produced by RoseTTAFold of two regions with the conserved cysteines of CyuA proteins. The first sequence and numbering correspond to MjCyuA.

**Funding:** This research received no external funding.

**Institutional Review Board Statement:** Not applicable.

**Informed Consent Statement:** Not applicable.

**Conflicts of Interest:** The author declares no conflict of interest.

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
