# Peer review of "Prediction of the Iron–Sulfur Binding Sites in Proteins Using the Highly Accurate Three-Dimensional Models Calculated by AlphaFold and RoseTTAFold"

_inorganics, doi:10.3390/inorganics10010002_

Round 1
Reviewer 1 Report
The research addresses the orientation of amino acids that presumably ligate the iron-sulfur cluster for several proteins. The conclusion is that the computational models are quite accurate, although one cannot be certain until obtaining an experimental model.
The research is original since it addresses the reliability of models produced by Alphafold and Rosettafold for iron-sulfur proteins by comparing it with experimental structures. It adds the comparison between computational models from state-of-the-art databases like Alphafold and experimental structures.
The authors could screen the databases for all iron-sulfur proteins (instead of several), and compare them with the corresponding experimental structure (if available). A comprehensive comparison of this class of proteins that reflects the RMSDs of the computational models relative to the experimental structures would be very interesting.
The conclusions are consistent with the evidence and addresses the main question and the references are appropriate.
Amino acid (and distance) labels in Figure 2a, 5a, 6 could be made larger. It is a little small.
The hyphens in Figure 1 seem to be distorted. They should be corrected before publication.
Line 272: 'especiallly' should be corrected to 'especially'.
Author Response
Reviewer 1
The research addresses the orientation of amino acids that presumably ligate the iron-sulfur cluster for several proteins. The conclusion is that the computational models are quite accurate, although one cannot be certain until obtaining an experimental model.
The research is original since it addresses the reliability of models produced by Alphafold and Rosettafold for iron-sulfur proteins by comparing it with experimental structures. It adds the comparison between computational models from state-of-the-art databases like Alphafold and experimental structures.
The authors could screen the databases for all iron-sulfur proteins (instead of several), and compare them with the corresponding experimental structure (if available). A comprehensive comparison of this class of proteins that reflects the RMSDs of the computational models relative to the experimental structures would be very interesting.
In fact, I missed an article published November 24th (just before submitting my article), Wehrspan, Z.J.; McDonnell, R.T.; Elcock, A.H. Identification of Iron-Sulfur (Fe-S) Cluster and Zinc (Zn) Binding Sites Within Proteomes Predicted by DeepMind's AlphaFold2 Program Dramatically Expands the Metalloproteome. J Mol Biol 2021, in press; DOI:10.1016/j.jmb.2021.167377
that screens the databases for all iron sulfur proteins using a newly developed ligand-search algorithm. Wehrspan et al. showed that, in a large number of cases out of the 362,198 AlphaFold models from the 21 complete AlphaFold proteomes screened by their ligand-search algorithm, the predicted [Fe-S] binding sites were sufficiently well-formed to bind the ligands. This paper nicely complements my article and answers the first part of the question. The reference and corresponding information from this paper have been added in the introduction. In addition, the conclusions of this article are compared to mine in the conclusion section.
However, concerning the second part of the question, it is very difficult to compare generally the computational models of [Fe-S]-proteins with the experimentally determined structures. If there is a close homologue in the PDB, of course the 3D models of Alphafold and Rosettafold of the protein with unknown structure will be very reliable and the FeS binding site will be predicted very accurately. And if there is no close homologue (the cases investigated in my article), the crystal structure remain to be solved to know exactly how correct the models are.
The conclusions are consistent with the evidence and addresses the main question and the references are appropriate.
Minor modifications
Amino acid (and distance) labels in Figure 2a, 5a, 6 could be made larger. It is a little small.
The hyphens in Figure 1 seem to be distorted. They should be corrected before publication.
Line 272: 'especiallly' should be corrected to 'especially'.
All minor points have been corrected

Reviewer 2 Report
The author evaluated the quality of models generated by the most state-of-the-art techniques for protein structure prediction, i.e., alphafold and rosettafold. She raised a question on the accuracy of the models for at least several iron-sulfur cluster binding proteins and suggests some limitations and applicability of the programs. I agree that there are definitely some limits in alphafold in respect to the accuracy and applicability, but these kinds of limitations are appeared to be generally hindered by their remarkable success stories. So I think her work is a very interesting and meaningful study at this point. The manuscript is well written and nicely describes the details of the key functional aspects of these proteins as well as the models' limitations to get these structural features.
I'd just like to ask the author about the following questions.
- It seems that all the residues contributing to cluster formation are gathered together and facing each other, so if it is predicted to this extent, the model can be considered quite accurate?
- The prediction of the disulfide seems to be quite challenging. I think even in the rosettafold model of BlsK (in Fig 6C) if the distance between two Cysteines is "a little bit" far away, they would not make a disulfide bond. I wonder if the interpretation of the model (at least the current version of the alphafold) should be considered in this way.
